# Genome-Wide Identification of the BTB Domain-Containing Protein Gene Family in Pepper (*Capsicum annuum* L.)

**DOI:** 10.3390/ijms26073429

**Published:** 2025-04-06

**Authors:** Qiaoling Yuan, Jin Wang, Feng Liu, Xiongze Dai, Fan Zhu, Xuexiao Zou, Cheng Xiong

**Affiliations:** 1Key Laboratory for Vegetable Biology of Hunan Province, Engineering Research Center for Horticultural Crop Germplasm Creation and New Variety Breeding, Ministry of Education, College of Horticulture, Hunan Agricultural University, Changsha 410125, China; qiaoling_y1993@163.com (Q.Y.); jinwang214@163.com (J.W.); jwszjx@hunau.edu.cn (F.L.); xiongzedai@126.com (X.D.); zf1968001@hunau.edu.cn (F.Z.); 2Yuelushan Lab, Changsha 410128, China

**Keywords:** *Capsicum annuum* L., *CaBTB* gene family, abiotic stresses, expression patterns, RT-qPCR

## Abstract

Pepper (*Capsicum annuum* L.), recognized as a globally preeminent vegetable, holds substantial economic and nutritional value. The BTB (broad-complex, tramtrack, and bric-a-brac) family of proteins, characterized by a highly conserved BTB domain, also denoted as the POZ domain, are intricately involved in a diverse array of biological processes. However, the existing corpus of research regarding pepper *BTB* genes remains relatively meager. In this study, a total of 72 *CaBTB* gene members were meticulously identified from the entire genome of pepper. Phylogenetic analysis illuminated the presence of conspicuous collinear relationships between the *CaBTB* genes and those of its closely affiliated species. Gene expression profiling and RT-qPCR analysis revealed that multiple *CaBTB* genes exhibited pronounced differential expression under diverse treatment regimens. Expression pattern analysis unveiled that *CaBTB25* manifested a remarkably elevated abundance in leaves. Moreover, its promoters were replete with an abundance of light-responsive *cis*-elements. Our comprehensive and in-depth explorations into subcellular localization revealed that CaBTB25 was predominantly detected to localize within the nucleus and lacked transcriptional activation. This research provides a crucial theoretical edifice, enabling a more profound understanding of the biological functions of the *BTB* gene family in pepper, thereby underscoring its potential significance within the intricate network of gene–environment interactions.

## 1. Introduction

The BTB (broad-complex, tramtrack, and bric-a-brac) protein was initially discovered in the fruit fly (*Drosophila melanogaster*). It encompasses a highly conserved domain consisting of approximately 115 amino acids [1]. Structurally, its core is composed of five *α*-helices (designated from A1 to A5) and three *β*-sheets (from B1 to B3). The A1/2 and A4/5 segments form an *α*-helical hairpin configuration, while the B1/B2/A1/A2/B3 region is connected to the A4/A5 region via helix A3 and a variable linker region [2]. Besides the core BTB domain, the diverse BTB subclasses feature extended regions at the amino-terminal (N-terminal) and carboxyl-terminal (C-terminal). These extensions are pivotal in endowing the protein with its specific functionality [2,3]. Based on the additional regions associated with the BTB domain, it can be classified into nine distinct subfamilies: BTB-only, BTB-BACK, BTB-Kelch, BTB-back-Kelch, BTB-Arm, BTB-TAZ, BTB-ANK, BTB-PHR, and Rho-BTB [4]. Currently, within the BTB protein family, research on the NPH3 subfamily is relatively inadequate. Previous studies suggest that NPH3 subfamily proteins may be involved in plant growth and development. In maize, the *BTB* family genes exhibit distinct tissue-specific expression patterns. Although the precise function of the NPH3 subfamily remains ill defined, it can be surmised that, akin to other subfamilies, they may partake in specific physiological processes during maize growth and development [5]. Moreover, studies have demonstrated that BTB-NPH3 is associated with phototropism, root geotropism, and auxin-mediated plant development [6]. The MATH subfamily, frequently encountered in proteins engaged in cytoplasmic signal transduction, and *BTB* genes bearing certain other functional domains, may potentially influence processes such as early flowering, branching, leaf development, and long-horned fruit development [7]. The BTB Armadillo (ARM) protein, characterized by an N-terminal ARM repeat sequence and a C-terminal BTB domain, may be implicated in the growth, development, and stress resistance of sugar beet [8]. However, its exact function necessitates further in-depth exploration. The BTB-TPR protein is distinguished by an N-terminal BTB domain and six TPR motifs, along with a coiled-coil motif at the C-terminal. This protein is exclusive to the plant kingdom. In *Arabidopsis thaliana*, it is represented by three members, namely *ETO1* and *EOL1/2*, which are involved in ethylene biosynthesis [9]. Proteins containing the BACK (BTB and C-terminal Kelch) domain comprise a BTB domain and a C-terminal Kelch repeat sequence. The BTB Ankyrin protein family is present in metazoans and higher plants. Its members possess an N-terminal BTB domain, a C-terminal ankyrin protein repeat domain, and a nuclear localization sequence [10].

BTB proteins have been extensively studied in the context of plant growth and development, stress resistance, protein ubiquitination and degradation, cytoskeleton organization, ion channel regulation, and cell cycle control [11,12]. Besides, they can interact with E3 ligase CUL3 to form an E3 ubiquitin ligase complex, thereby participating in the ubiquitination process of target proteins [13]. For instance, an interaction between OsMBTB32, a protein encompassing a MATH-BTB domain, and OsCUL1s has been detected to modulate salt tolerance in *Oryza sativa* [14]. The BTB-BACK-TAZ domain-containing protein MdBT2 negatively modulates the drought tolerance of apple plantlets by attenuating the positive regulatory effect of MdHDZ27 through ubiquitination [15]. Four *CmBTs* genes negatively regulate root development in chrysanthemum by mediating nitrate assimilation, amino acid biosynthesis, and the auxin and jasmonic acid (JA) signaling pathways [16]. Additionally, OsBTB97 harbors two BBX proteins, and the OsBBX19-OsBTB97/OsBBX11 module may be involved in spikelet development and seed production of rice [17].

Multiple members of the BTB protein family are involved in regulating the reaction of the plant-related signal network system mediated by hormones such as SA, JA, ABA, GA_3_, etc. Subsequently, they regulate a series of physiological and biochemical processes, including plant germination, metabolism, growth, and development [3,18,19,20]. Yeast two-hybrid (Y2H) and bimolecular fluorescence complementation (BiFC) assays reveal that the IbBT4 protein interacts with BR-ENHANCED EXPRESSION 2 (BEE2) to enhance the BR signaling pathway and proline biosynthesis and further activate the ROS-scavenging system. Ultimately, the drought resistance of transgenic *Arabidopsis thaliana* is augmented [21]. A comprehensive RT-qPCR analysis of 11 poplar *BTB* genes in leaf, root, and xylem tissues indicates their expression patterns in response to different hormonal and biotic/abiotic stress conditions, with varying degrees of regulation manifested in the results [22]. In soybean, multiple *GmBTBs* are verified by RT-qPCR to be significantly upregulated under salinity, drought, and nitrate stress conditions [23]. Studies have shown that *CaBPM4*-silenced plants exhibit markedly increased peroxidase activity and decreased malondialdehyde concentrations, suggesting that *CaBPM4* may enhance resistance to salt and drought stress in pepper [24]. In summary, BTB proteins play an essential role in regulating plant stress responses.

Pepper (*Capsicum annuum* L.) represents a vegetable crop of substantial economic importance on a global scale. Nevertheless, the extant literature regarding the members of the *BTB* genes has been rather scarce heretofore. Consequently, in this study, a genome-wide exploration was undertaken, entailing comprehensive bioinformatics and expression analyses of *CaBTBs*, with the aim of attaining a more profound comprehension of the pepper *CaBTB* genes. Specifically, a total of 72 members of the *CaBTB* family were identified. Subsequently, their phylogenetic relationships, chromosomal localizations, gene architectures, and conserved motifs were meticulously elucidated. The expression profiles in diverse tissues and under a plethora of stress conditions were also scrutinized. Notably, *CaBTB25* was significantly enriched in leaves and harbored an abundance of light-responsive *cis*-elements within its promoters, strongly suggesting its involvement in the process of photomorphogenesis. In summary, this research expounds upon the evolutionary trajectory and functional divergence of the *BTB* gene family in pepper. Moreover, it proffers potential gene resources for the genetic enhancement of pepper, particularly with respect to tolerance to light and stress.

## 2. Results

### 2.1. Identification of BTB Genes in Pepper

In reference to the sequences of BTB proteins in *Arabidopsis thaliana* and *Oryza sativa*, a total of 72 *CaBTB* genes were identified within the ‘Zunla-1’ genome databases. These genes were renamed as *CaBTB1* to *CaBTB72*, contingent upon their distributions and relative linear orders along their respective chromosomes (Appendix A). The analysis outcomes disclosed that the divergent genomic characteristics, coding sequence lengths, and amino acid sequences culminated in distinct relative molecular masses (MW) and isoelectric points (pI) among the identified *CaBTB* candidates. The lengths of the 72 putative proteins spanned from 126 (*CaBTB15*) to 966 (*CaBTB53*) amino acid residues, with predicted molecular weights (MW) ranging from 14.1 to 110.1 kDa. Moreover, the isoelectric points (pI) of the *CaBTB* gene family exhibited a spectrum from 3.88 to 9.87, with an average value of 6.44. A multitude of CaBTB proteins were localized in the nucleus, endomembrane system, and plasma membrane, while a small number were found in the chloroplast and other subcellular compartments.

### 2.2. Phylogenetic Analysis of BTB Family

To further delineate the evolutionary pathway of *BTB* genes across diverse species, a phylogenetic tree was constructed based on 72 pepper BTB amino acid sequences, along with those from tomato and *Arabidopsis* (Figure 1). The bootstrap values, which ranged predominantly from 0.88 to 1, demonstrated a reliable phylogenetic relationship. The graphical illustration revealed that all the identified BTB proteins were classified into seven subfamilies, comprising NPR1 (nonexpressor of pathogenesis-related proteins 1), MATH (meprin and TRAF homology), C-terminal Kelch-related, TAZ zinc finger, ARM (Armadillo/beta-catenin-like repeat), SKP1 (S-phase kinase-associated protein 1), and NPH3 (Nonphototropic hypocotyl 3), respectively. Notably, the NPH3 domain was the most prevalent putative domain. It was specifically present in pepper and *Arabidopsis*, with the exception of SlBTB13 of tomato.

### 2.3. Structure Characterization and Motif Composition of CaBTB Genes

To comprehensively understand the structural diversity, similarity, and evolutionary relationships of *CaBTB* genes, an analysis was conducted on conserved motifs, protein structures, and exon–intron arrangements. Using the MEME suite, ten conserved motifs were identified among members of the *CaBTB* gene family. Each gene contained 1–7 motifs (Figure 2a,b). The sequence logos of these motifs, presented in Appendix A, had a minimum length of 21 amino acids (motif 3) and a maximum length of 80 amino acids (motif 1). Given that members of the same subfamily shared similar motifs, the 72 *CaBTB* genes were classified into four clades. Notably, except for *CaBTB47*, motif 8 was unique to clade I at the C-terminal and absent in other clades, suggesting its significance in conferring specialized genetic functions. CaBTB proteins were categorized into several subfamilies based on the domains at the N- and C-terminal, such as the BTB-NPH3 subfamily (Figure 2c). The gene structure of *CaBTB* gene family members was characterized by the exon–intron pattern (Figure 2d). The results indicated that within the same subfamily, almost all *CaBTB* gene structures were highly conserved. However, the variation in exon–intron structures among the four distinct subgroups contributed to the diversification of the functional range of the *CaBTB* gene family during evolution.

### 2.4. Chromosomal Localization and Duplication Analysis of CaBTB Genes

Chromosome mapping analysis revealed that approximately 94% of *CaBTB* family genes were heterogeneously distributed across 12 chromosomes. However, four genes (*CaBTB69*, *CaBTB70*, *CaBTB71*, and *CaBTB72*) were located on as-yet-unmapped scaffolds (Figure 3). Among the chromosomes, chromosome 2 harbored the largest number of *CaBTB* genes, followed by chromosomes 1, 6, 7, and 9, respectively. Chromosome 11 had the fewest *CaBTB* genes. Interestingly, a significant number of genes were clustered at the chromosome ends, while the middle regions contained a relatively small number of genes. Segmental duplications analysis, following MCScanX searching and manual screening, confirmed the existence of *CaBTB* gene pairs. Moreover, most segmental duplications occurred between different pepper chromosomes, with fewer within the same chromosome. These findings were highly consistent with the evolutionary grouping tree, as each pair of segmentally duplicated genes was located within the same phylogenetic clade.

### 2.5. Cis-Acting Element Prediction of CaBTB Promoters

Investigating the putative *cis*-acting elements is crucial for a deeper familiarization of the *CaBTB* gene expression level. Additionally, it helps elucidate the potential functions and molecular mechanisms under the regulation of their corresponding trans-regulatory factors. Therefore, the promoter sequences located 2000 bp upstream of *CaBTB* genes were retrieved using TBtools and analyzed via the PlantCARE database for *cis*-acting element identification. The data revealed the presence of a total of 47 *cis*-acting elements. Invariably, the core promoter elements TATA-box and CAAT-box (not shown in the legend due to their large numbers) were present in all *CaBTB* promoters. The remaining *cis*-acting elements were mainly classified into four categories (Figure 4 and Appendix A). Light-responsive elements were the most prevalent, amounting to 16 (34%), indicating that *CaBTB* gene family members are highly induced by light. Among them, another category consisted of 12 elements that functioned as abiotic and biotic stress-responsive elements, including box-S, CARE, CCAAT-box, LTR, MBS, MYB, MYC, STRE, TC-rich repeats, W box, WRE3, and WUN-motif. Specifically, 10 elements were associated with the adaptation to endogenous hormone changes: ABRE, as-1, AuxRR-core, CGTCA-motif, DRE, ERE, GARE-motif, TCA-element, TGACG-motif, and TGA-element. Moreover, the remaining nine *cis*-acting elements, accounting for 19% of the total number of elements, were involved in plant growth and development. Collectively, the presence of diverse response components provides empirical support for the regulatory role of *CaBTB* genes in coordinating plant biotic and abiotic responses through hormone signaling pathways.

### 2.6. Expression Pattern Analysis of CaBTB Genes in Various Tissues

To precisely determine the transcription levels of 72 pepper *BTB* genes across 14 tissues, namely root, stem, leaf, bud, flower, and nine stages of fruit development, a hierarchical clustering heatmap was constructed based on the previously obtained ‘Zunla-1’ transcriptome data. In view of the aforementioned *cis*-element analysis findings, the majority of *CaBTB* genes are presumably involved in the photoperiodic response. Given that the principal site of photoperiod perception in plants is the leaf, especially the mature leaf, six *CaBTB* genes (*CaBTB1*, *CaBTB21*, *CaBTB25*, *CaBTB29*, *CaBTB34*, and *CaBTB42*) exhibiting relatively high transcriptional levels in leaves were selected for RT-qPCR validation to corroborate the accuracy of the data. The RT-qPCR results were congruent with those of RNA-sequencing (Figure 5b–g). Among the identified *CaBTB* genes, *CaBTB3*, *CaBTB14*, and *CaBTB72* were clearly not expressed. Conversely, the vast majority of genes were expressed across nearly all tissues and developmental stages. Nevertheless, several genes demonstrated tissue-specific and stage-dependent expression patterns (Figure 5a). For instance, *CaBTB32*, *CaBTB60*, and *CaBTB70*, members of clade IV, exhibited pronounced enrichment in buds and floral tissues. This expression profile strongly suggests that these genes play pivotal roles in the morphogenesis and ontogeny of floral organs.

### 2.7. Expression Profiles of CaBTB Genes in Response to Exogenous Hormones

The *CaBTB* gene family members likely partake in multiple biological processes, contingent upon the profusion of *cis*-acting elements enriched within the promoter analysis outcomes. These elements are associated with phytohormone responses and abiotic stress. To explore the potential functions of *CaBTB* genes in their response to phytohormones, heat map visualizations were employed to analyze the expression profiles of all *CaBTB* genes in either pepper leaf or root tissues. These tissues were subjected to five phytohormone treatments—ABA, GA_3_, IAA, JA, and SA—and were contrasted with untreated control samples. The results revealed that the expression levels of the majority of *CaBTB* genes were markedly influenced by diverse treatments, manifesting as varying degrees of upregulation or downregulation. Nevertheless, *CaBTB3*, *CaBTB14*, *CaBTB56*, and *CaBTB66* were not detectable in terms of expression, thereby elucidating the co-existence of both correlations and disparities in the expression patterns among these *CaBTB* members.

Figure 6a demonstrated that upon treatment with abscisic acid (ABA, 30 μM), the expression levels of *CaBTB7*, *CaBTB30*, *CaBTB36*, *CaBTB48*, and *CaBTB72* were upregulated in leaves at stage AL4. Notably, *CaBTB28* was distinctly upregulated only at stage AL2. Conversely, *CaBTB10*, *CaBTB18*, *CaBTB69*, and *CaBTB71* showed pronounced downregulation in expression at stage AL3. In contrast, the expression of three other *CaBTB* genes, namely clustered *CaBTB32*, *CaBTB51*, and *CaBTB70*, was significantly upregulated in roots. When treated with gibberellic acid (GA_3_, 2 μM), there was a substantial difference in expression between leaves and roots. Specifically, *CaBTB27*, *CaBTB36*, *CaBTB37*, *CaBTB51*, and *CaBTB67* were significantly upregulated in leaves, while the expressions of *CaBTB49* and *CaBTB63* were inhibited by GA_3_ treatment (Figure 6b). In response to indole acetic acid (IAA, 2 μM) treatment, the majority of *CaBTB* gene expressions were downregulated in leaves, with a particularly significant decrease in the transcriptional abundance of the *CaBTB6* gene. Notably, *CaBTB50* and *CaBTB65* showed markedly higher upregulation when exposed to IAA treatment in roots (Figure 6c). Similarly, after jasmonic acid (JA, 10 μM) treatment, numerous gene expressions were downregulated. For instance, *CaBTB2*, *CaBTB8*, *CaBTB21*, and *CaBTB49* genes underwent significant changes (Figure 6d). However, *CaBTB36*, *CaBTB37*, and *CaBTB58* were upregulated in leaves. Simultaneously, most pepper *BTB* genes were upregulated in roots under JA treatment. During the salicylic acid (SA, 2 mM) treatment, the expression levels of *CaBTBs* showed a significant response in roots, presenting an opposite pattern to that in leaves (Figure 6e).

### 2.8. Expression Profiles of CaBTB Genes Under Multiple Abiotic Stresses

To further explore the potential role of pepper *BTB* genes under multiple abiotic stresses, we analyzed their expression patterns. Pepper leaves and roots were treated with low-temperature stress (cold), oxidative stress (H_2_O_2_), high-temperature stress (heat), dehydration stress (mannitol), and salt stress (NaCl). Low-temperature treatment significantly induced the upregulation of *CaBTB57* and *CaBTB65* in leaves (Figure 7a) while downregulating *CaBTB24*, *CaBTB39*, *CaBTB49*, and *CaBTB63*. The expression patterns of some genes (*CaBTB3*, *CaBTB14*, *CaBTB15*, *CaBTB28*, *CaBTB56*, and *CaBTB66*) were undetectable. In roots, cold stress led to a significant increase in the expression levels of *CaBTB1*, *CaBTB27*, *CaBTB38*, *CaBTB57*, and *CaBTB58*, while *CaBTB2* and *CaBTB48* were markedly suppressed. The gene expression changes under H_2_O_2_ treatment were found to be similar to those under low-temperature stress (Figure 7b). Notably, *CaBTB27* and *CaBTB57* showed a more pronounced increase in leaves with enhanced oxidation effects. Elevated temperature mainly upregulated the expression of *CaBTB51* and *CaBTB72* in both leaves and roots (Figure 7c). Conversely, *CaBTB4* and *CaBTB42* were sharply downregulated by high temperature. Almost all *CaBTB* genes showed opposite expression patterns in leaves and roots under drought stress (Figure 7d). For example, *CaBTB47* was upregulated in leaves but downregulated in roots. In leaves treated with sodium chloride stress (Figure 7e), *CaBTB2*, *CaBTB10*, *CaBTB18*, *CaBTB53*, and *CaBTB71* were transcribed at relatively low levels, while *CaBTB7*, *CaBTB16*, *CaBTB30*, *CaBTB36*, *CaBTB37*, *CaBTB47*, and *CaBTB58* were evidently enriched. In roots, except for the increased expression of *CaBTB51* and *CaBTB65*, the expression changes of other genes were not significant.

### 2.9. Homology, Transcription Activation, and Subcellular Localization of CaBTB25

In this study, our focus was directed towards the *CaBTB25* gene. Given its particularly high expression in leaves and the presence of a greater number of light-responsive *cis*-elements in its promoter regions, it was hypothesized that *CaBTB25* might play a pivotal role in photomorphogenesis. To further explore this, a phylogenetic tree was constructed by comparing the amino acid sequences of CaBTB25 with its homologous sequences from 29 other species. As depicted in Figure 8a, CaBTB25 exhibited the closest relationship with tobacco (*Nicotiana tabacum* L.) due to their membership of the same Solanaceae family. Moreover, the growth status of yeast cells was examined to assess the transcriptional activity of CaBTB25. Using the combinations BD-Lam + AD-T and BD + AD as negative controls and the combination BD-53 + AD-T as the positive control, it was revealed that the full-length CaBTB25 lacked transcriptional activity (Figure 8b). Subcellular localization analysis of the CaBTB25-eGFP fusion protein, carried out in *N. benthamiana* leaves with mCherry serving as a marker for nuclear proteins, demonstrated that CaBTB25 was localized within the nucleus (Figure 8c).

## 3. Discussion

As previously documented, proteins belonging to the BTB/POZ family play pivotal roles in a plethora of functions within plants, encompassing growth and development, cytoskeletal modulation, chromatin remodeling, transcriptional regulation, and proteolytic degradation [7]. Extensive research has been carried out on BTB proteins in animals [25,26,27]. In recent times, BTB proteins have been identified in a wide array of plant species, with 158 members in rice [28], 38 members in tomato [29], 95 members in poplar [22], 122 members in soybean [23], 69 members in grape [30], and 62 members in *paulownia* [31]. In this study, we executed a comprehensive and systematic bioinformatics-based analysis of the BTB family members in chili pepper. The number of *CaBTB* genes was determined to be 72, which reveals the distinct genomic sizes across diverse plant species, suggesting that *BTB* genes have undergone evolutionary alterations. In this investigation, based on the BTB domain in conjunction with other conserved domains, 72 *CaBTB* members were categorized into 7 subfamilies. However, in comparison to the tomato *BTB* subfamily classification, pepper lacks the KCTD subfamily (Figure 1). This phenomenon may be attributed to the specific gene contraction in pepper, which has adapted to specific ecological environments during long-term evolution. Additionally, the completeness and accuracy of genome annotation have, to some degree, influenced the results.

Based on the phylogenetic relationships, the *CaBTB* genes in pepper were classified into four distinct clades comprising seven subfamilies. Members within these subfamilies exhibited similarities in terms of their motifs, protein structures, and exon–intron organizations. In peach, four novel subgroups of BTB proteins—BTBAND, BTBBL, BTBP, and BTBAN—were identified for the first time [32]. Notably, the newly discovered BTBAND subgroup was demonstrated to be a composite of ANK, NPR, and DUF domains. These domains, which are components of these new subgroups, have been detected in previous investigations [2,4,29], suggesting that these genes may execute novel or hitherto uncharacterized functions during plant growth and development. These studies imply that within the diverse subfamilies of BTB proteins, each subfamily member likely performs analogous functions in plant-specific biological processes. There exist intricate interactions between proteins and other molecules within different subcellular compartments. Elucidating the subcellular localization of proteins is essential for uncovering the underlying mechanisms of these molecular interactions. The expression patterns and subcellular localization of BPM proteins in *Arabidopsis* display distinct characteristics across various tissues and under abiotic stress conditions [33]. By investigating gene structure and motif distribution, the composition, organization, and evolution of genes in different organisms can be ascertained. This knowledge forms the cornerstone for comprehending gene function and regulatory mechanisms. We also observed a high degree of conservation among BTB family members in pepper, resulting in genes within the same subfamily sharing identical or similar exon–intron structures and motif types (Figure 2). Evidently, genes with multiple introns may experience transcriptional retardation, and their gene expression levels may be inhibited in response to stressful environments [34,35]. The motifs within the BTB domain are ubiquitously present across eukaryotes and are significant for their interactions with other proteins, transcriptional activity, and DNA binding [36]. In pepper *BTB* genes, we identified 10 motifs, which may aid in the identification of potential transcription factor binding sites, thus facilitating our understanding of the diverse gene regulatory mechanisms (Appendix A).

The *CaBTB* genes exhibit an uneven distribution across the twelve chromosomes of the pepper genome. The overwhelming majority of these genes are localized at the termini of the chromosomes and predominantly exist in tandem duplications (Figure 3). Gene duplication represents one of the primary driving forces in genome evolution. In pepper, gene duplication events are likely responsible for the expansion of the *BTB* gene family. A substantial number of gene duplications provide pepper with supplementary genetic material and augment gene diversity [37]. For example, in other plant species, gene duplication events frequently precipitate the expansion of gene families, endowing plants with the capacity to adapt to diverse environmental conditions. As a crucial economic crop, the expansion of the *BTB* gene family in pepper may facilitate its survival and reproduction in various growth environments. In grasses, the MATH-BTB protein functions as a substrate-specific adaptor within the context of ubiquitin E3 ligase and exerts a regulatory effect during the transition from meiosis to mitosis. In contrast to *Arabidopsis*, which possesses only six homologous genes, this family has undergone extensive amplification in grasses. Research has shown that this amplification is predominantly attributed to local gene duplication [38].

Analyzing the spatiotemporal expression patterns of genes, along with the expression levels of regulated genes and their responses to environmental and endogenous signals, can reveal promoter analysis [39,40]. This offers novel insights and methodologies for researching the molecular mechanisms of plant growth and development, breeding superior varieties, and studying plant development. In sweet potato, several stress-responsive *cis*-acting regulatory elements, including the GARE-motif, O2-site, LTR, TC-rich repeats, TCA-element, MBS, ARE, and MBSI, were detected in the promoter regions of *IbBT4* [21]. In *GmBTBs* promoters, various light-, hormone-, and stress-responsiveness *cis*-elements were identified, suggesting that the associated proteins may be involved in soybean growth and stress responses [23]. Regarding the *cis*-elements analysis in *CaBTB* gene promoters, we obtained a series of abiotic and biotic stress-responsive elements, light-responsive elements, phytohormone-responsive elements, and plant growth and development-responsive elements (Figure 4 and Supplemental Figure 2). Furthermore, light-responsive elements are ubiquitously present and are the most abundant, which is consistent with the findings in tomato [29], sugar beet [8], grape [30], and *paulownias* [31]. This implies that BTB family members are likely influenced by light in most plant species. It is essential to highlight that the ubiquitin ligase CONSTITUTIVELY PHOTOMORPHOGENIC 1 (COP1) assumes a pivotal role in light-induced anthocyanin biosynthesis. In the context of apple, MdBT2, a BTB protein, has been demonstrated to directly interact with and stabilize MdCOP1 through the inhibition of self-ubiquitination [41]. This process is closely intertwined with the photoperiodic response, as light perception and subsequent signaling pathways are fundamental to regulating anthocyanin production. The potential connection between *CaBTB* genes and the COP1-mediated, light-induced anthocyanin biosynthesis pathway warrants further investigation to elucidate the complex molecular mechanisms underlying photoperiod-related physiological processes in plants. Within the purview of this study, *CaBTB25*, the CaBTB25-eGFP fusion protein of which is localized to the nucleus and devoid of transcriptional activity, manifested the most conspicuous level of expression in leaves, and its promoter was replete with an ample quality of light-responsive *cis*-acting elements (Figure 8). It is postulated that this phenomenon is associated with photomorphogenesis. Meanwhile, stress-responsive elements such as MBS, MYB, and W box were also identified in the promoters of *CaBTB* genes. Previous research on pepper has demonstrated that the transcript level of *CaBPM4* (*CaBTB62*) increases when the plant is exposed to abiotic stresses such as salt, cold, and drought, owing to the presence of corresponding response elements in the promoter site [24]. In conclusion, the analysis of pepper *cis*-elements is of great significance for uncovering gene expression regulation mechanisms, studying pepper’s response to adversity, guiding pepper variety improvement, and exploring the functions of the pepper gene family.

Based on the expression patterns and RT-qPCR results of *CaBTB* genes in diverse tissues, our findings demonstrated that three genes, namely *CaBTB3*, *CaBTB14*, and *CaBTB72*, were not expressed. Conversely, the remaining genes exhibited high expression levels across nearly all the examined pepper tissues (Figure 5a). Certain specific *CaBTB* genes, such as *CaBTB1*, *CaBTB21*, *CaBTB25*, *CaBTB29*, *CaBTB34*, and *CaBTB42*, manifested high expression levels in the leaves, implying their potential involvement in leaf development (Figure 5b–g). TMF (TERMINATING FLOWER) is recognized as a transcription factor during the formation of multiflowered inflorescences. In tomato, BOP (BLADE-ON-PETIOLE) transcriptional cofactors, characterized by the conserved BTB/POZ domain, were found to mediate the regulation of inflorescence architecture through interaction with TMF [42]. Intriguingly, *CaBTB60*, which is a homologous gene of *SlBOP3*, is highly transcribed in pepper buds and flowers. This suggests that *BTB* genes may play a role in flower development, although their specific functions await further investigation.

Previously, studies have elucidated that the BTB protein assumes a critical role in plants’ defense mechanisms against both abiotic and biotic stresses. A multitude of plant BTB/POZ protein family members are implicated in governing hormone-mediated processes and phytologically relevant signal network responses [31]. These proteins serve as pivotal regulators, precisely modulating the intricate signaling cascades initiated by diverse hormones, including JA [43], SA [44,45], ABA [46], and GA_3_. *OsBTBZ1*, a salt-tolerant gene operating within the ABA-dependent pathways, not only restores the phenotypes of the *bt3* mutant line but also augments the growth of wild-type (WT) *Arabidopsis* under stress conditions when ectopically overexpressed [46]. Additionally, in the ethylene (ET)/jasmonic acid (JA) pathway of *Arabidopsis thaliana*, the expression of *AtBT4* is markedly upregulated, thereby promoting the plant’s resistance to *Botrytis griseus* [47]. The DELLA protein functions as a central regulatory factor that interconnects the gibberellin (GA) signaling pathway with other signaling pathways. It accomplishes this by directly interacting with key transcription factors or regulators, thus playing an indispensable role in integrating and coordinating various signaling processes within the plant [48,49]. Research findings demonstrate that MdBT2 is involved in regulating nitrate-induced plant growth through its interaction with the DELLA protein MdRGL3a. Furthermore, compared to WT plants, the OE-MdBT2 lines exhibit a substantial increase in plant height and biomass [50]. In our study, to explore the potential response capabilities of *CaBTB* genes to abiotic and biotic stresses, we harnessed RNA-seq data to analyze their expression patterns under cold, H_2_O_2_, heat, mannitol, and NaCl conditions, as well as various hormone treatments. Obviously, the expression level of *CaBTB36*, which is homologous to *MdBT2* in apple, was remarkably upregulated upon treatment with ABA, GA_3_, and JA. These results imply that *CaBTB36* may play significant roles in plant responses to different environmental stimuli and hormonal regulations. Further research on its function could offer deeper insights into the molecular mechanisms underlying plant stress responses and growth regulations. By modulating these processes, BTB proteins can exert an influence on plant growth, development, and responses to environmental stimuli. Their functions in signal network responses are also of utmost importance for plants to coordinate different physiological processes, ensuring proper growth and survival in changing environments. This further underscores the central and diverse roles of the BTB protein family in plant biology.

## 4. Materials and Methods

### 4.1. Plant Materials and Growth Conditions

In this study, the wild-type (WT) parent ‘Zhangshugang’ line, a widely employed backbone and breeding parent in China’s pepper breeding program and an advanced inbred line, was sourced from the Pepper Research Laboratory, College of Horticulture, Hunan Agricultural University (latitude 28°11′49″ N, longitude 112°58′429″ E, Changsha, China). Prior to cultivation, seeds were germinated at 55 °C for 15 min. Subsequently, all plants were cultivated in growth chambers under a 16-h light (25 °C)/8-h dark (18 °C) photoperiod cycle, with a relative humidity of 65% and a light intensity of 300 µmol·m^−2^·s^−1^. In the context of *Nicotiana tabacum* L., nuclear-localized tobacco proteins were tagged with a nuclear-localization marker. Concurrently, tobacco plants were cultured in an incubator maintained at a constant temperature of 22 °C. The photoperiod was strictly regulated to a 16-hour light/8-hour dark cycle. At the 8-to-10-leaf developmental stage, leaves that were fully expanded and exhibited optimal physiological vigor were meticulously selected for subsequent biochemical analyses.

### 4.2. Identification of Pepper Genes Encoding BTB Domain Proteins

The CaBTB protein sequences were retrieved from the high-quality ‘Zunla-1’ Pepper Genomic Database website (https://solgenomics.net/organism/Capsicum_annuum/genome, accessed on 13 January 2025). The *Solanum lycopersicum* L. genome (SGN: http://solgenomics.net/, accessed on 5 January 2025) and *Arabidopsis thaliana* genome (TAIR: https://www.Arabidopsis.org/, accessed on 5 January 2025) were referenced. Moreover, the HMM file (PF00651) of the conserved domain was obtained from the Pfam database website (http://pfam-legacy.xfam.org/, accessed on 13 January 2025). The prediction of the BTB domain was validated using the NCBI-CDD database (http://www.ncbi.nlm.nih.gov/cdd/, accessed on 13 January 2025) and SMART database (http://smart.embl-heidelberg.de, accessed on 13 January 2025), with redundant sequences being filtered out. The physicochemical properties, such as the relative molecular weight (MW) and isoelectric point (pI) of the final CaBTB proteins, were analyzed via the ExPASy website (http://www.expasy.org, accessed on 13 January 2025). Subcellular localization prediction was conducted using the BUSCA website (https://busca.biocomp.unibo.it/, accessed on 13 January 2025) and CELLO online analysis tool (http://cello.life.nctu.edu.tw/, accessed on 13 January 2025).

### 4.3. Phylogenetic Analysis of the BTB Proteins in Pepper

In the context of analyzing the amino acid sequences of tomato (*Solanum lycopersicum*) and *Arabidopsis thaliana*, in addition to those of pepper (*Capsicum annuum*) BTB family members, the MEGA (Molecular Evolutionary Genetics Analysis, available at (https://www.megasoftware.net/, Version 11.0)) software was utilized to construct a phylogenetic tree. The neighbor-joining (NJ) algorithm was implemented, with the *p*-distance model adopted as the substitution model and 1000 bootstrap replicates performed for statistical robustness. As a result, the *CaBTB* gene family in pepper was partitioned into four distinct subfamilies.

### 4.4. Gene Structure and Significant Motifs Analysis

TBtools (Version 2.152) was harnessed to delineate the gene structure maps of *CaBTB* gene family members [51]. Concurrently, the exon–intron distribution was annotated based on the coding DNA sequence (CDS) and genome sequences of the *CaBTB* genes, leveraging the GSDS (http://gsds.cbi.pku.edu.cn/, accessed on 13 January 2025) visualization platform. For the prediction of conserved motifs, the Multiple Expectation Maximization for Motif Elucidation (MEME) tool (http://meme-suite.org/tools/meme, accessed on 13 January 2025) was utilized. The search mode was configured as ‘any number of repetitions’, the number of motifs was set to 10, the minimum width was specified as 6 amino acid residues, and the maximum width was set at 100 amino acid residues.

### 4.5. Chromosomal Distribution and Detection of Gene Duplication

The chromosomal localization of each *CaBTB* gene was visualized by discerning its position on chromosomes, referring to the online Sol Genomics Network (SGN) database. The MCScanX (Version 2019) software was utilized to detect segmental duplication alignments and identify intraspecific gene pairs within the pepper genome. Subsequently, the chromosome mapping and gene duplication results of the *CaBTB* genes were visualized using TBtools software, enabling a comprehensive and detailed display of the genetic relationships and distribution patterns.

### 4.6. Prediction of the Cis-Acting Elements in the CaBTB Promoters

The 2-kilobase (kb) upstream regions of each CDS within the *CaBTB* gene family were extracted using TBtools software, which were designated as the promoter regions. Subsequently, all the *BTB* genes under investigation were submitted to the PlantCARE online database (http://bioinformatics.psb.ugent.be/webtools/plantcare/html/, accessed on 15 January 2025) for the prediction of primary stress-response *cis*-elements. Finally, all the identified *cis*-acting elements were visualized with the assistance of TBtools software, facilitating a comprehensive and intuitive understanding of their distribution patterns.

### 4.7. Expression Pattern Analysis of Pepper CaBTB Genes

The gene expression and RNA-seq data were downloaded from the National Center for Biotechnology Information (NCBI) under the accession number GSE45037 and the Pepper Information Data Center (http://lifenglab.hzau.edu.cn/PepperHub/index.php, accessed on 15 January 2025) [52,53]. The ‘Zunla-1’ genome was utilized as the reference genome. Experimental treatment conditions and statistical analysis methods were implemented in accordance with those described in previous studies [53]. Subsequently, the gene expression results were calculated, and heatmaps were generated using TBtools software, enabling a comprehensive visualization and interpretation of the gene expression profiles.

### 4.8. RNA Isolation and Reverse Transcription Quantitative PCR (RT-qPCR)

Total RNA was extracted from the leaves of WT pepper ‘Zhangshugang’ using the FastPure^®^ Universal Plant Total RNA Isolation Kit (Vazyme Biotech Co., Ltd., Piscataway, NJ, USA). Each sample group was prepared with three biological replicates. Subsequently, 1 μg of the isolated total RNA was reverse-transcribed into complementary DNA (cDNA) using the HiScript^®^ IIQ RT SuperMix (+ gDNAwiper) kit (Vazyme Biotech) in a 20 μL reaction system. Real-time quantitative polymerase chain reaction (RT-qPCR) was carried out on a LightCycler^®^ 96 Real-Time PCR System (Roche, Basel, Switzerland) with a 20 μL reaction system, employing the ChamQ Universal SYBR qPCR Master Mix (Vazyme Biotech Co., Ltd., Piscataway, NJ, USA). The relative expression levels were calculated using the 2^−ΔΔCt^ method [54], with *CaActin7* (*Caz04g16060*) serving as the internal reference gene for normalization. The specific primers utilized in this study are listed in Appendix A. These primers were designed using the GenScript Online PCR Primers Designs Tool (https://www.genscript.com/tools/pcr-primers-designer, accessed on 15 January 2025).

### 4.9. Subcellular Localization

The full-length open reading frames (ORFs) of *CaBTB25*, excluding the stop codon, were inserted into the pSuper1300-eGFP plasmid via homologous recombination, thereby generating the CaBTB25-eGFP fusion protein construct. Subsequently, this recombinant plasmid was introduced into *Agrobacterium* competent GV3101 (Weidi Biotechnology Co., Ltd., Shanghai, China). The correct bacterial solution was first cultured in LB medium supplemented with kanamycin and rifampicin antibiotics overnight for 16 h. Then, 1 mL of the sample was transferred and further cultured in 20 mL of induced medium (comprising 50 μg·mL^−1^ kanamycin, 50 μg·mL^−1^ rifampicin, 10 mM MES, and 20 μM acetosyringone [AS]) for an additional 12 h. The bacterial liquid was then collected by centrifugation at 5000 rpm for 10 min. The supernatant was discarded, and an appropriate volume of infection buffer was added for resuspension. The infection buffer consisted of 10 mM MES and 10 mM MgCl_2_, with the pH adjusted to 5.6. Subsequently, AS was added to reach a final concentration of 200 μM. Finally, the optical density at 600 nm (OD_600_) of the resuspended bacterial solution was adjusted to 1.0. This prepared bacterial suspension was then inoculated into the nuclear localization *Nicotiana benthamiana* leaves. The inoculated plants were incubated at 28 °C in the dark for at least 3 h before injection. Then, 2–3 days post-injection, the optimal time for observation, the enhanced green fluorescence protein was visualized using a laser confocal microscope (Zeiss LSM510 META, Oberkochen, Germany) and photographed. The primer sequences used in this experimental procedure are presented in Appendix A.

### 4.10. Transcriptional Activation Analysis

The full-length coding sequence (CDS) of *CaBTB25* was amplified from the cDNA extracted from the young and tender leaves of the ‘Zhangshugang’ pepper variety. The amplified product was then integrated into a plasmid vector, and the reconstructed plasmid was subjected to sequencing to confirm its integrity and accuracy. Subsequently, in accordance with the corresponding instruction manual, the plasmid was transformed into Y2HGold chemically competent cells (Weidi Biotechnology Co., Ltd., Shanghai, China). Following transformation, the bacterial strain was cultured on SD/-Trp-Leu (synthetic dropout medium lacking tryptophan and leucine) at 30 °C for a duration of three days. This selective medium was used to screen for cells that had successfully taken up the plasmid. Positive clones were then selected and further cultivated on SD/-Trp-Leu-His-Ade (synthetic dropout medium lacking tryptophan, leucine, histidine, and adenine) for a period of 3–5 days, with appropriate controls being maintained concurrently. The primer sequences utilized in this experimental procedure are detailed in Appendix A.

## 5. Conclusions

In our research, an exhaustive exploration of the pepper (*Capsicum annuum* L.) genome culminated in the successful discernment of 72 *CaBTB* genes, which were subsequently subjected to an intensive analysis. This analysis encompassed diverse facets, such as their physicochemical attributes, phylogenetic interrelationships, subcellular localization prognoses, gene architectures, duplication occurrences, and promoter *cis*-regulatory elements. The BTB domain was phylogenetically categorized into seven discrete subfamilies. A substantial number of *CaBTB* genes manifested tissue-specific expression profiles across diverse tissues. Furthermore, they evinced responsiveness to five phytohormone regimens, specifically abscisic acid (ABA), gibberellic acid (GA_3_), indole-3-acetic acid (IAA), jasmonic acid (JA), and salicylic acid (SA), as well as five abiotic stress treatments, namely cold stress, hydrogen peroxide (H_2_O_2_) stress, heat stress, mannitol-induced osmotic stress, and sodium chloride (NaCl)-induced salinity stress. The results from RT-qPCR incontrovertibly demonstrated substantial upregulation or downregulation of numerous *CaBTB* genes in response to specific hormonal and stress stimuli, thereby furnishing robust experimental substantiation for the responsiveness of the *CaBTB* gene family to these incitements. Notably, the *CaBTB25* gene was screened as a candidate gene that could be regulated by light. In summary, the genome-wide identification and characterization of the *BTB* gene family in pepper are pivotal in unearthing the potential applications of these research outcomes in pepper breeding initiatives and agricultural practices.

## Figures and Tables

**Figure 1 ijms-26-03429-f001:**
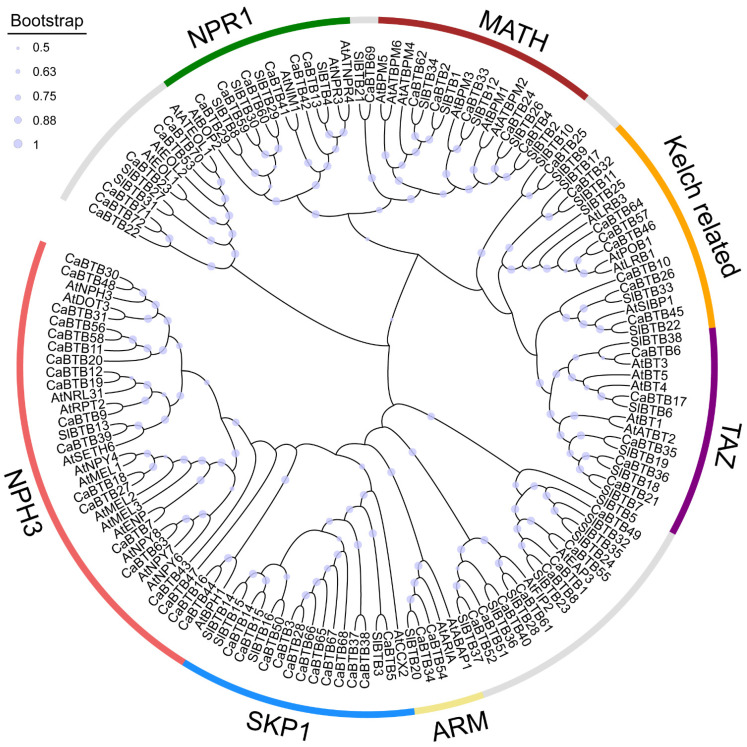
Phylogenetic analysis of the *BTB* gene family in *Capsicum annuum*, *Solanum lycopersicum*, and *Arabidopsis thaliana* was conducted. The full-length sequences of BTB proteins were meticulously analyzed to construct an evolutionary tree. To this end, the MEGA5.2 software was employed, and the neighbor-joining algorithm was implemented with 1000 bootstrap replications to ensure the robustness and reliability of the phylogenetic inferences.

**Figure 2 ijms-26-03429-f002:**
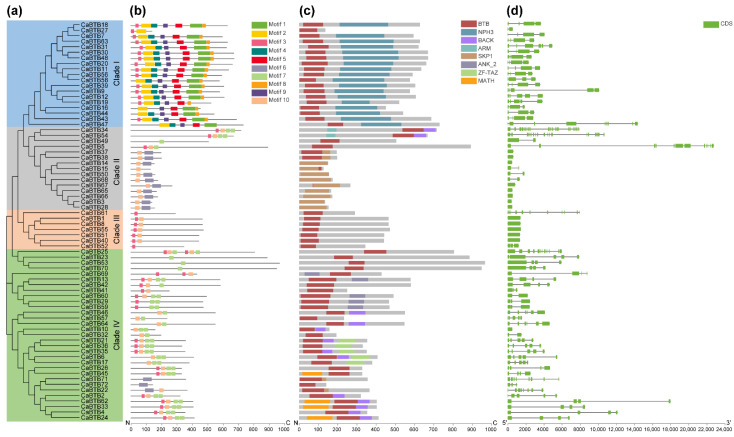
Conserved motif and gene structure analysis of *CaBTBs*. (**a**) Phylogenetic relationship of *CaBTB* gene family. The *CaBTB* gene family in pepper was classified into four distinct clades according to its evolutionary relationship. (**b**) Gene motifs analysis. A total of ten conserved motif structures were identified in CaBTB proteins. These motifs are represented by diverse-colored rectangles, which visually display the distribution of motifs among different CaBTB proteins. (**c**) Structure of the pepper BTB proteins. The *CaBTB* gene family was discriminated based on the functional domains located at the N- and C-terminal of the BTB domain (**d**) Gene structure. The green block represents the CDS, and the solid line indicates the intron.

**Figure 3 ijms-26-03429-f003:**
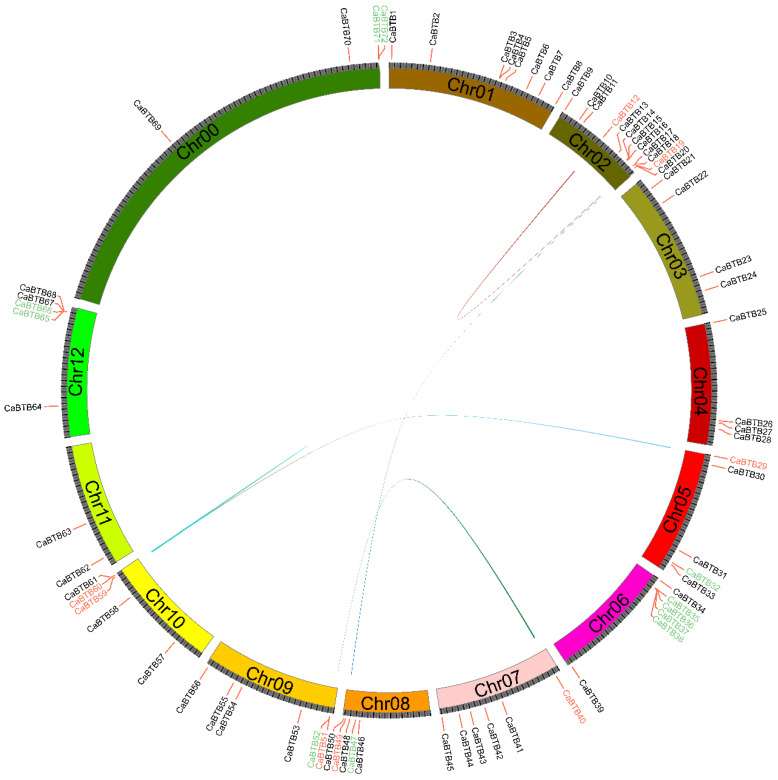
Chromosomal localization and duplication events of *BTB* genes in pepper. The decentralized rings denote the distribution of consecutive chromosomes and the interchromosomal linkages. Multicolored solid-dotted lines are applied to signify the duplicating gene pairs of *CaBTBs*.

**Figure 4 ijms-26-03429-f004:**
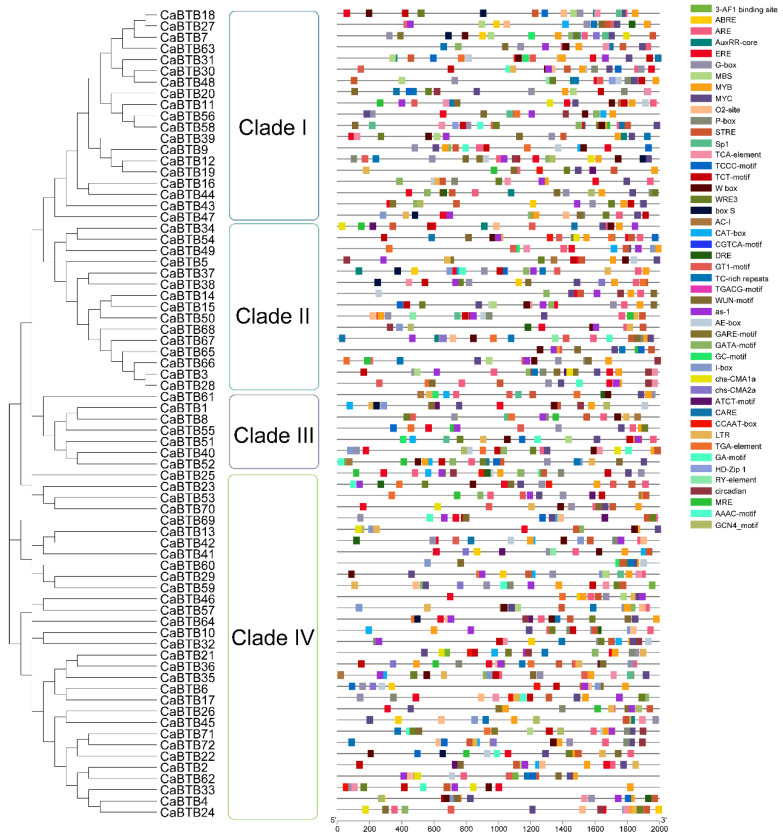
Analysis of *cis*-acting element in the *BTB* gene family of pepper.

**Figure 5 ijms-26-03429-f005:**
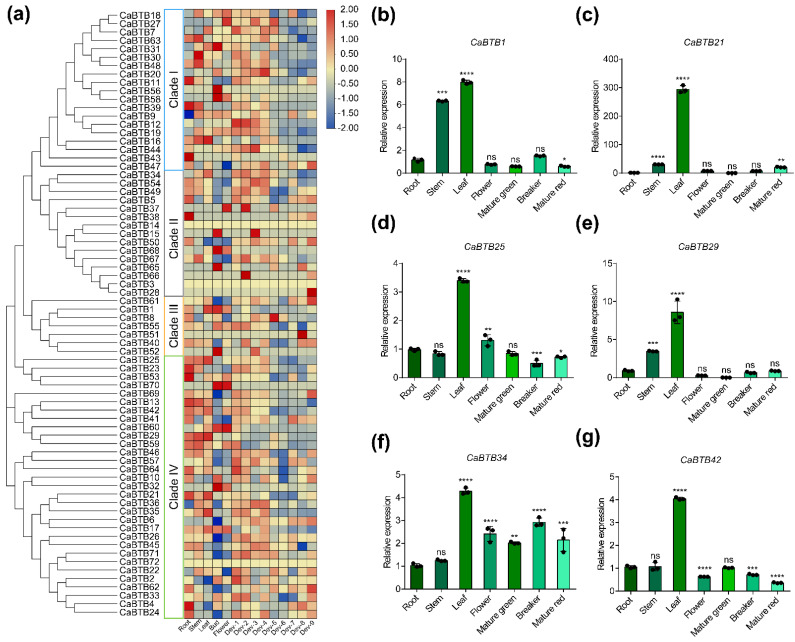
Expression pattern of *CaBTB* genes in pepper. (**a**) By integrating the outcomes of phylogenetic analysis, we harnessed transcriptome datasets to comprehensively profile the tissue-specific expression patterns of the *CaBTB* gene family across diverse tissues and organs. The tissues include the following: root (developing roots), stem (developing stems), leaf (mature leaves), bud (closed flower buds), flower (blooming flowers), Dev1 (fruits with a length of 0–1 cm), Dev2 (fruits with a length of 1–3 cm), Dev3 (fruits with a length of 3–4 cm), Dev4 (fruits with a length of 4–5 cm), Dev5 (mature green fruits), Dev6 (fruits at breaker stage), Dev7 (fruits 3 days after breaker stage), Dev8 (fruits 5 days after breaker stage), and Dev9 (fruits 7 days after breaker stage). (**b**–**g**) The relative expression levels of six *CaBTB* genes in various tissues were measured. Values are presented as means ± SD (*n* = 3 biological replicates). Significant differences were determined by one-way ANOVA analysis (* *p* < 0.05; ** *p* < 0.01; *** *p* < 0.001; **** *p* < 0.0001; ns, no significant difference).

**Figure 6 ijms-26-03429-f006:**
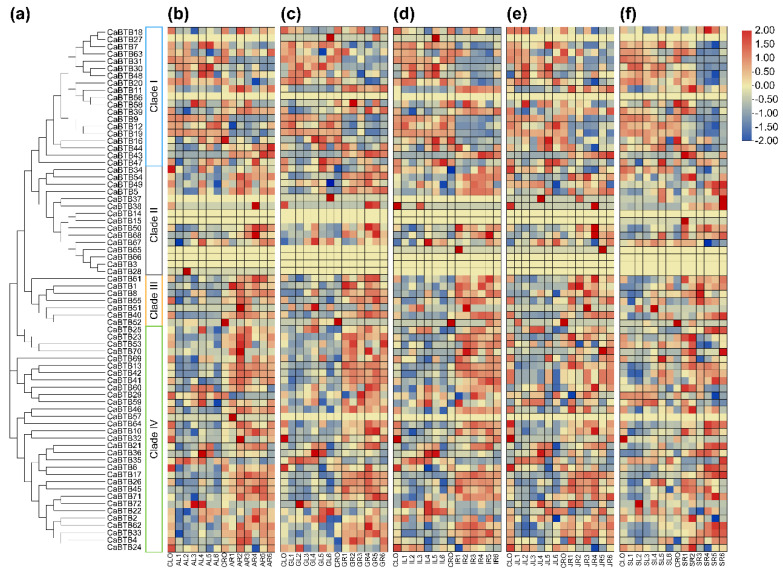
Expression profiles of *CaBTB* genes under exogenous hormones. Red blocks denote genes with high expression levels, while blue blocks represent genes with low expression levels. (**a**) Phylogenetic analysis. (**b**) ABA treatment. (**c**) GA_3_ treatment. (**d**) IAA treatment. (**e**) JA treatment. (**f**) SA treatment. CL: control leaf; CR: control root; AL: ABA-treated leaf; AR: ABA-treated root; GL: GA_3_-treated leaf; GR: GA_3_-treated root; IL: IAA-treated leaf; IR: IAA-treated root; JL: JA-treated leaf; JR: JA-treated root; SL: SA-treated leaf; SR: SA-treated root. All samples were collected from leaves or roots, with three biological replicates for each treatment.

**Figure 7 ijms-26-03429-f007:**
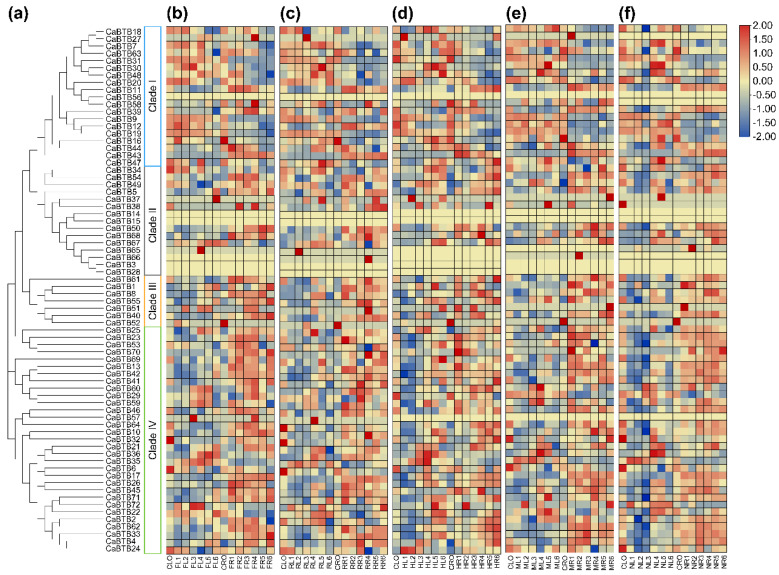
Expression profiles of *CaBTB* genes under different abiotic stresses. Colorful squares indicate log_2_ expression value. (**a**) Phylogenetic analysis. (**b**) Cold treatment. (**c**) H_2_O_2_ treatment. (**d**) Heat treatment. (**e**) Mannitol treatment. (**f**) NaCl treatment. CL: control leaf; CR: control root; FL: cold-treated leaf; FR: cold-treated root; RL: H_2_O_2_-treated leaf; RR: H_2_O_2_-treated root; HL: heat-treated leaf; HR: heat-treated root; ML: mannitol-treated leaf; MR: mannitol-treated root; NL: NaCl-treated leaf; NR: NaCl-treated root. All samples were collected from leaves or roots, with three biological replicates for each treatment.

**Figure 8 ijms-26-03429-f008:**
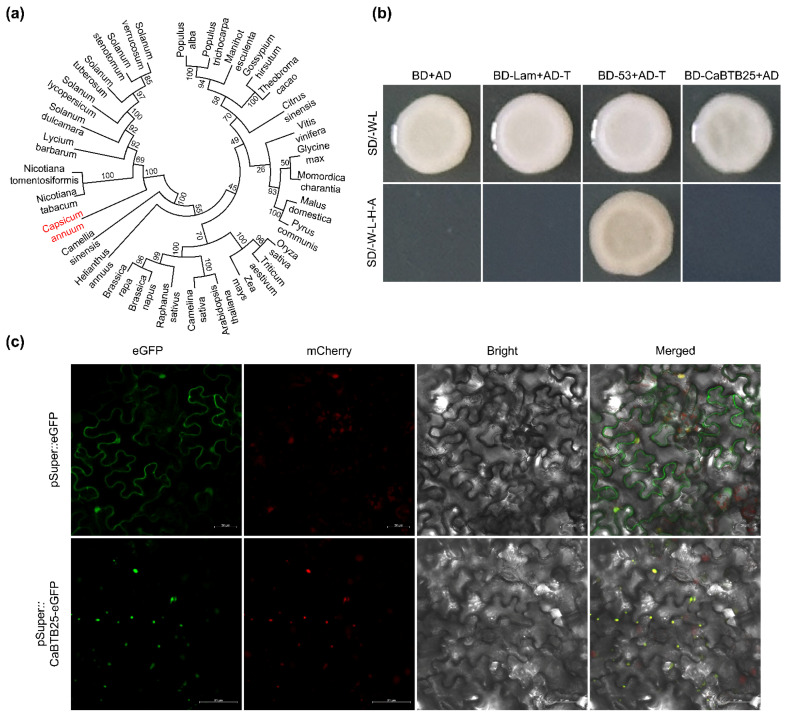
Gene basic information of *CaBTB25*. (**a**) Phyloenetic analysis of CaBTB25 with its homolog proteins in other species. (**b**) Transcriptional activation of CaBTB25 in yeast cells. BD + AD indicates pGBKT7 + pGADT7; BD-Lam + AD-T indicates pGBKT7-Lam + pGADT7-T; BD-53 + AD-T indicates pGBKT7-53 + pGADT7-T; BD-CaBTB25 + AD indicates pGBKT7-CaBTB25 + pGADT7. SD/-W-L, selective medium lacking tryptophan and leucine; SD/-W-L-H-A, selective medium lacking tryptophan, leucine, histidine, and adenine. (**c**) Subcellular localization analysis of CaBTB25-eGFP recombinant protein. Scale bar, 50 μm.

## Data Availability

All data are contained within the article and Appendix A.

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
