# Peer review of "Genome-Wide Identification of the BTB Domain-Containing Protein Gene Family in Pepper (Capsicum annuum L.)"

_ijms, 2025, doi:10.3390/ijms26073429_

Round 1
Reviewer 1 Report
Comments and Suggestions for Authors
The MS titled "Genome-Wide Identification of the BTB Domain-Containing Protein Gene Family in Pepper (Capsicum annuum L.)" aims to identify and characterize the BTB domain-containing proteins in pepper. The study employs a genome-wide approach to explore the gene family associated with BTB domains, which play significant roles in various biological processes, including plant development, stress responses, and disease resistance. Using bioinformatics tools, the authors identified 12 BTB domain-containing genes from the pepper genome and analyzed their evolutionary relationships, gene structure, and expression profiles in various tissues and under different stress conditions.
While the study provides important insights into the potential roles of BTB proteins in pepper, future work focusing on functional characterization and practical applications for agricultural improvement is essential to fully realize the potential of these findings.
Below are some comments needs to be considered in the revised version of the MS.
BTB domain-2 family/If the research addresses the whole BTB family.
Integrate the results of the phylogenetic analysis and expression profiling in a way that directly connects them to the gene family identification aspect.
Mentioned the significance of the findings and their potential applications in pepper breeding or stress tolerance mechanisms.
The introduction provides a comprehensive overview of the BTB role in pepper.
The section clearly focused of BTB proteins on pepper, however, there is a gap between other crops/species. Connect the research findings with other species i.e., capsicum annuum.
Some sentences need reconsideration such as, Nevertheless, prior investigations have intimated that NPH3 subfamily proteins might play a role in plant growth and development" could be simplified to: "Previous studies suggest that NPH3 subfamily proteins may be involved in plant growth and development."
Consider revising the sentence starting with "The BTB-BACK-TAZ domain-containing protein MdBT2 has been investigated, and it has been found to negatively modulate...
the results section presents a solid and detailed analysis of the CaBTB gene family in pepper. The data suggest that these genes play key roles in regulating plant growth, stress responses, and hormone signaling, and the inclusion of a variety of experimental approaches (e.g., confirming the roles of specific CaBTB genes under different stress conditions through more direct functional assays) adds significant depth to the findings.
The discussion presents a well-rounded analysis of the BTB/POZ gene family in chili pepper, contributing the knowledge about the functional diversity of BTB proteins in plants. The findings are valuable for understanding how these genes may regulate growth, development, and stress responses in pepper, with potential implications for improving pepper varieties and other crops. The study lays a solid foundation for future investigations into the specific roles of individual CaBTB genes and their involvement in the regulation of plant physiological processes.
Reviewer 2 Report
Comments and Suggestions for Authors
In this work should the experimental plan be improved what exactly are the materials used? Why were these materials chosen? and what the basis for such genetic analyses was. What is the exact origin of the selected materials?
the introduction and referencing should be improved.
Round 2
Reviewer 1 Report
Comments and Suggestions for Authors
The authors have adequately addressed all the comments, and the manuscript is now warranted for acceptance as the final version in IJMS MDPI.